# Divergent and gram-scale syntheses of (−)-veratramine and (−)-cyclopamine

Wenlong Hou[1], Hao Lin[1], Yanru Wu[1], Chuang Li[1], Jiajun Chen[1], Xiao-Yu Liu [1]✉ & Yong Qin [1]✉

Veratramine and cyclopamine, two of the most representative members of the isosteroidal alkaloids, are valuable molecules in agricultural and medicinal chemistry. While plant extraction of these compounds suffers from uncertain supply, efficient chemical synthesis approaches are in high demand. Here, we present concise, divergent, and scalable syntheses of veratramine and cyclopamine with 11% and 6.2% overall yield, respectively, from inexpensive dehydro-*epi*-androsterone. Our synthesis readily provides gram quantities of both target natural products by utilizing a biomimetic rearrangement to form the C-*nor*-D-*homo* steroid core and a stereoselective reductive coupling/(bis-)cyclization sequence to establish the (E)/F-ring moiety.

The isosteroidal alkaloids (ISAs, e.g., **1**–**7**; Fig. 1a) are characteristic chemical components from the plants of the genera *Veratrum* and *Fritillaria*[1,2]. Architecturally, the ISAs feature a common C-*nor*-D-*homo* steroid skeleton and could be classified into three structural types (i.e., veratramine, jervanine, and cevanine; Fig. 1a) based on the connectivity of the parent A/B/C/D core and the piperidine F ring[2]. These unique molecules exhibit various biological activities, including significant analgesic, anticancer, antitussive, and insecticidal effects[1–7]. In particular, the representative member of the veratramine-type alkaloids, veratramine (**1**), has been used as a commercialized biopesticide in China[8]. In addition, the jervanine alkaloid cyclopamine (**4**) was identified as a potent inhibitor of the Hedgehog signaling pathway[9,10], which has resulted in the developing of a new class of cancer therapeutics[11]. Of note, a semisynthetic analog of **4**, patidegib [SGT-610, previously named saridegib (IPI-926)][12,13], is currently in phase III clinical trial for the treatment of basal cell nevus syndrome (Gorlin syndrome) (https://clinicaltrials.gov/study/NCT06050122?intr=Patidegib&rank=1).

The chemical and biological significances of ISAs render them attractive targets in the synthetic community[14–25]. So far, seven ISAs belonging to the aforementioned three skeletal types have been synthesized by worldwide chemists (Fig. 1a). We were intrigued by the veratramine- and jervanine-type ISAs, the previous syntheses of which are summarized in Fig. 1b. Specifically, the groups of Johnson[14,15] and Masamune[16] achieved pioneering total syntheses of veratramine (**1**) and jervine (**3**) in 1967, which proceeded in 44 and 47 steps from

Hagemann's ester (**8**), respectively. In 1968, Kutney et al. published a 34-step synthetic approach to verarine (**2**) from 6-methoxy-2-tetralone (**9**)[17]. Over 40 years later, Giannis et al. reported an elegant semi-synthesis of cyclopamine (**4**) in 26 steps from dehydro-*epi*-androsterone (DHEA, **10**)[20]. Very recently in 2023, Baran's group disclosed a convergent total synthesis of cyclopamine (**4**) starting from Wieland-Miescher ketone (**11**) in 16 steps (longest linear sequence, LLS) and 22 total steps[23]. Immediately after this work, Gao and colleagues accomplished a concise synthesis of veratramine (**1**) from **11** in 13 steps (LLS, 23 total steps) and an eight-step relay synthesis of cyclopamine (**4**) from **1**[24]. Collectively, all prior syntheses afforded the veratramine- and jervanine-type ISAs with over 22–47 total steps and in milligram (mg) scale from commercially available materials. Despite these advances, innovative synthetic access to both two types of ISAs remains highly desirable. As our long-standing interest in the syntheses of fascinating alkaloids[26–28], herein, we report divergent, efficient, and gram-scale syntheses of (−)-veratramine (**1**) and (−)-cyclopamine (**4**) in 13 steps (LLS, 15 total steps) from inexpensive DHEA (**10**, ca. $0.25/gram)[29].

Our retrosynthetic analysis of veratramine (**1**) and cyclopamine (**4**) is outlined in Fig. 1c. Structurally, the target C-*nor*-D-*homo* steroid alkaloids **1** and **4** are common in possessing a piperidine F ring, but different in bearing a disconnected E ring and aromatic D ring in the former and a spiro tetrahydrofuran E ring in the latter. We envisioned assembly of the F ring in **1** at a late stage via a reductive coupling/cyclization sequence (**12** + **14** to **A** to **1**) from imine **12** and aldehyde **14**.

[1]Key Laboratory of Drug-Targeting and Drug Delivery System of the Education Ministry and Sichuan Province, Sichuan Engineering Laboratory for Plant-Sourced Drug and Sichuan Research Center for Drug Precision Industrial Technology, West China School of Pharmacy, Sichuan University, Chengdu, China. ✉e-mail: xyliu@scu.edu.cn; yongqin@scu.edu.cn

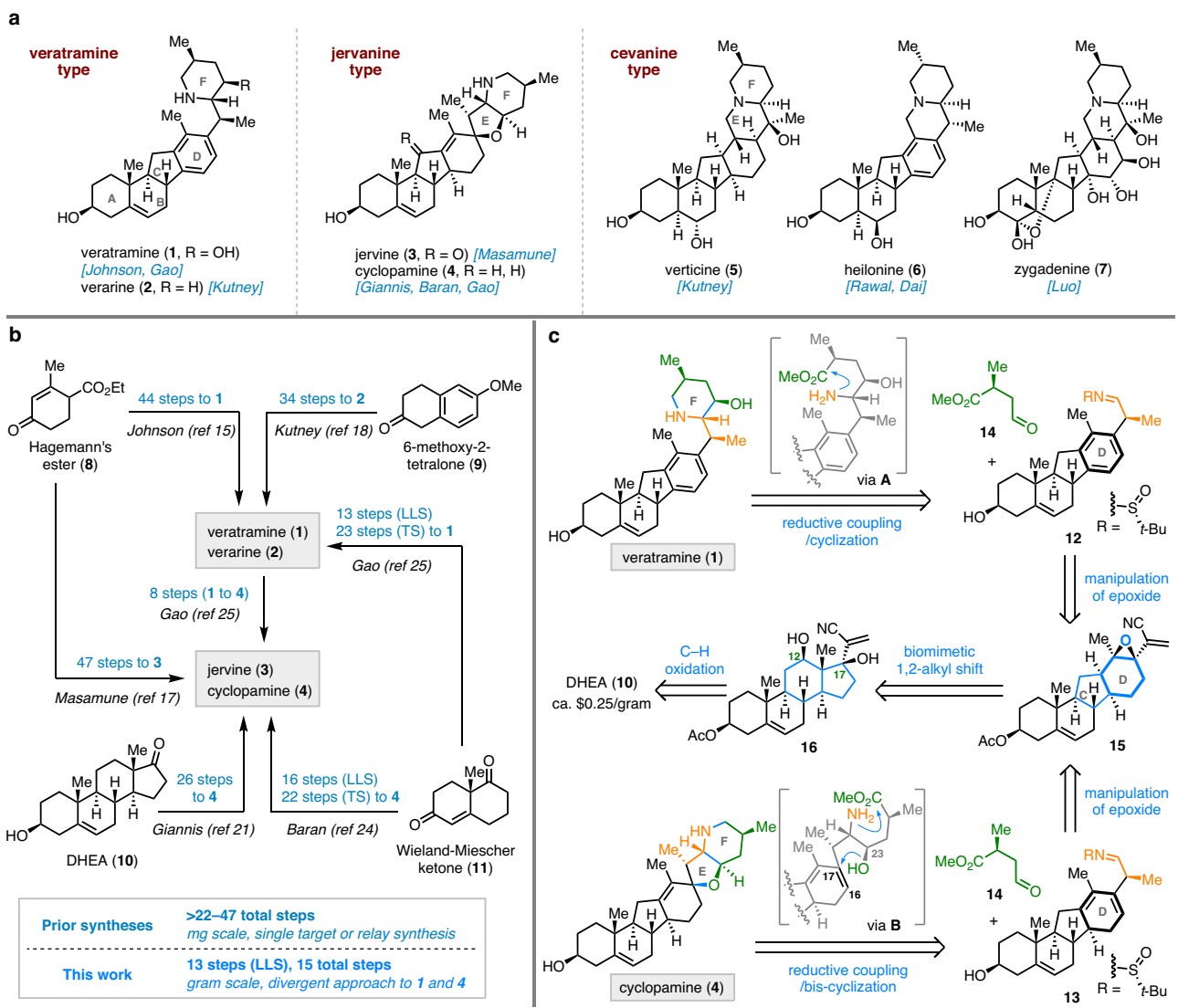

**Fig. 1 | Background and study synopsis. a** Structures and chemical syntheses of three types of ISAs. **b** Summary of the prior syntheses of veratramine- and jervanine-type ISAs and an overview of this work. **c** Retrosynthetic analysis of veratramine (**1**) and cyclopamine (**4**). ISAs, isosteroidal alkaloids; LLS, longest linear sequence; TS, total steps.

Notably, the same strategy could be employed for simultaneously setting up the E/F rings in **4** through reductive coupling between imine **13** and aldehyde **14**, followed by bis-cyclization of intermediate **B** including 1) from the C23 hydroxyl group onto C16–C17 alkene and 2) from the resulting amine onto the carboxylic ester. The key stereogenic centers of the β-amino alcohol moiety in **1** and **4** would be secured by the chiral *tert*-butylsulfinamide group in the reductive coupling reaction[30,31]. Both **12** and **13** could be traced backed to the common precursor **15** through divergent manipulations of the *tetra*-substituted epoxy group to generate an arene D ring (in **12**) and a cyclohexadiene D ring (in **13**), respectively. A cyano group in **15** was envisaged for the formation of the *N*-sulfinyl imine in **12** and **13**. In turn, the advanced intermediate **15** would be formed by a biomimetic 1,2-alkyl shift from **16** to construct the C-*nor*-D-*homo* steroid framework. Finally, **16** could be prepared through C12 oxidation[32–34] and C17 addition from DHEA (**10**).

## Results

### Biomimetic rearrangement to form the C-*nor*-D-*homo* steroid core

In the forward synthesis (Fig. 2a), we first prepared the known compound **17**[32] from DHEA (**10**) based on a Cu-mediated C–H oxidation protocol, which was originally developed by Schönecker[33] and later improved by Baran, Garcia-Bosch, and co-workers[32,34]. Following the modified conditions[34], this process was easily carried out at a large scale and afforded decagram quantities of **17** (74% overall yield from **10**, see Supplementary Information, page S3). Grignard addition of ethynylmagnesium bromide to the C17 carbonyl group in **17** gave **18** (84% yield). The terminal alkyne **18** was subjected to the nickel-catalyzed hydrocyanation conditions [Ni(acac)$_2$, Mn, Zn(CN)$_2$, neocuproine] developed by Liu and co-workers[35], efficiently furnishing the expected vinyl nitrile **16** in 91% yield at 18 g scale. The installed cyano group at the C20 position proved to be important and served as an appropriate synthetic handle for the late-stage formation of the piperidine F ring (*vide infra*).

Next, we set out to investigate the key biomimetic C14(13→12) rearrangement to convert a typical 6/6/6/5 steroid into its 6/6/5/6 counterpart. While such a transformation has been documented in the literature to prepare C-*nor*-D-*homo* steroid[20,36–42], we speculated that pre-installation of a C17 hydroxyl group in **16** would not only facilitate the rearrangement but also act as a proper functionality to form the anticipated arene D ring or cyclohexadiene D ring for divergently accessing our target alkaloids. After screening of reaction conditions, the desired C14(13→12) rearrangement of diol **16** occurred in the

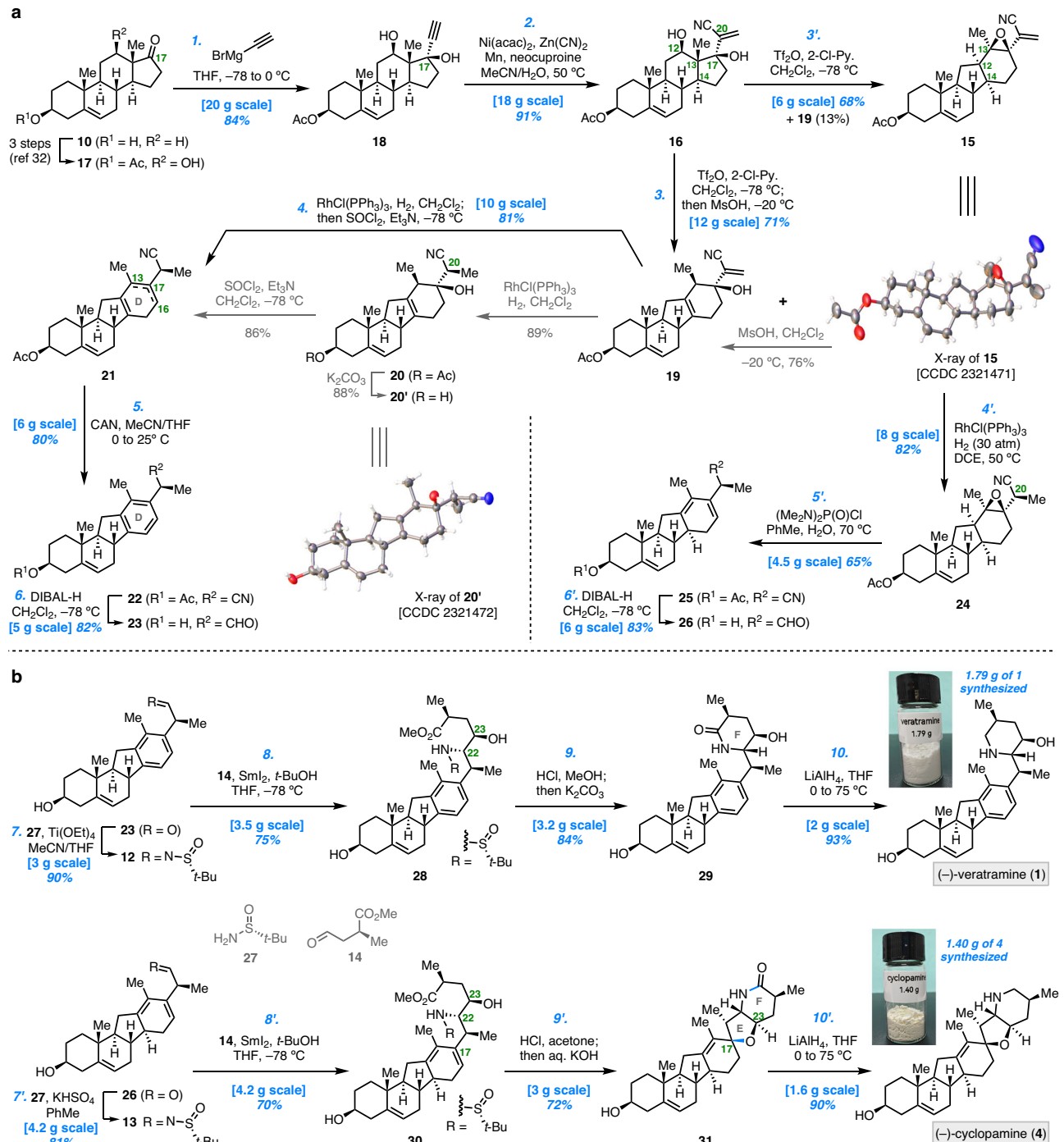

**Fig. 2 | Divergent syntheses of (−)-veratramine (1) and (−)-cyclopamine (4).** **a** Preparation of advanced intermediates **23** and **26** with C-*nor*-D-*homo* steroid core. **b** Construction of F ring in **1** via reductive coupling/cyclization and construction of E/F rings in **4** via reductive coupling/bis-cyclization. CAN, ceric ammonium nitrate; DIBAL-H, diisobutylaluminium hydride; neocuproine, 2,9-dimethyl-1,10-phenanthroline.

presence of trifluoromethanesulfonic anhydride (Tf₂O) and 2-chloropyridine (2-Cl-Py.) at − 78 °C, which afforded epoxide **15** (68% yield, 6 g scale) as the major product, along with alcohol **19** (13% yield). An X-ray crystallographic analysis of **15** unambiguously confirmed its 6/6/5/6 tetracyclic structure including the presence of an epoxy unit. Alcohol **19** could be generated from **15** by treating with methanesulfonic acid (MsOH), presumably through epoxide opening and subsequent 1,2-hydride shift and deprotonation (Supplementary Information, page S7). We were able to perform the aforementioned two transformations in a single step by directly adding MsOH to the

reaction mixture after completion of the rearrangement process, converting **16** into **19** in 71% yield on a 12 g scale. The above sequence rapidly delivered the rearrangement of products **15** and **19** suitable for elaboration to the desired D ring.

## Divergent manipulation of epoxide

With a reliable biomimetic rearrangement protocol in hand, we then explored divergent preparation of the advanced intermediates **12** and **13** according to the synthetic plan. While attempts on direct conversion of epoxide **15** or homoallylic alcohol **19** to arene **22** failed[43–45], an

**Fig. 3 | Syntheses of diverse C-*nor*-D-*homo* steroid analogues.** Substrate scope of the biomimetic rearrangement reaction.

alternative stepwise approach was adopted. Thus, regio- and diastereoselective reduction of the terminal olefin in **19** was accomplished via catalytic hydrogenation with Wilkinson's catalyst [RhCl(PPh₃)₃], giving the product **20** as a single diastereomer. Analysis of the X-ray structure of **20′**, a deacylated derivative of **20**, verified that the correct configuration at C20 was established. Of note, the directed hydrogenation through coordination of the rhodium catalyst with β-oriented O-atom of the hydroxyl group in **19** secured the correct C20 stereocenter[46]. Dehydration of tertiary alcohol **20** with SOCl₂ and Et₃N delivered **21** as an inseparable mixture of Δ¹³,¹⁷ and Δ¹⁶,¹⁷ isomers (ca. 1.5:1 ratio). Conducting the hydrogenation and dehydration in one step allowed efficient conversion of **19** into **21** in 81% yield at 10 g scale. Both olefin isomers in **21** were useful as they were subsequently transformed into arene **22** (80% yield) through oxidative aromatization of the D ring using ceric ammonium nitrate (CAN). On the other hand, subjecting alkene **15** to directed hydrogenation conditions employing Wilkinson's catalyst furnished **24** as the predominant product (d.r. = 33:1). The newly generated C20 configuration in **24** was determined as the desired one in the following transformations (*vide infra*). After screening different reaction conditions, we were delighted to observe the smooth conversion of epoxide **24** into the expected conjugated diene **25** (65% yield, 4.5 g scale) with tetramethyldiamidophosphoric acid chloride[45] and water in PhMe at 70 °C. Oxidative aromatization of diene **25** in the presence of CAN yielded **22** (Supplementary Information, page S22), which verified the correct C20 configuration.

### Syntheses of (−)-veratramine and (−)-cyclopamine

Having efficiently prepared **22** and **25**, our remaining tasks were to forge the F ring in veratramine (**1**) and the E/F rings in cyclopamine (**4**). Therefore, subjecting **22** to DIBAL-H in CH₂Cl₂ at −78 °C promoted reduction of the acyl and cyano groups simultaneously to furnish

aldehyde **23** in 82% yield. Condensation of aldehyde **23** with (*R*)-*tert*-butanesulfinamide (**27**) in the presence of Ti(OEt)₄ produced *N*-sulfinyl imine **12** (Fig. 2b). With gram quantities of **12** in hand, assembly of the piperidine F ring was investigated. As a result, the crucial reductive cross-coupling of *N*-*tert*-butanesulfinyl imine **12** with known aldehyde **14**[47] (prepared in two steps, see Supplementary Information, page S13) proceeded using SmI₂[30] and delivered amino alcohol **28** in 75% yield at 3.5 g scale (d.r. > 30:1 based on crude ¹H-NMR). Excellent control of the configurations at newly formed C22 and C23 stereocenters was induced by the chirality of the (*R*)-*tert*-butanesulfinamide motif (Supplementary Information, page S14). Removal of the sulfinyl group in **28** with HCl/MeOH solution, followed by treating the resultant amine with K₂CO₃, gave rise to **29** (84% yield) with F ring formed through lactamization. Ultimately, (−)-veratramine (**1**) was prepared from **29** via reduction of the amide group with LiAlH₄. The present 13-step synthesis readily provides **1** with 1.79 g quantities, which exhibits identical spectroscopic data with those of natural[48] and previously synthesized material[24].

Synthesis of cyclopamine (**4**) was completed following a similar reductive coupling/bis-cyclization strategy (Fig. 2b). Specifically, DIBAL-H reduction first converted nitrile **25** into aldehyde **26**. While condensation of **26** with (*R*)-*tert*-butanesulfinamide (**27**) was ineffective in the presence of Ti(OEt)₄, the reaction could be improved using KHSO₄, which gave *N*-sulfinyl imine **13** in 81% yield. SmI₂-mediated reductive coupling of imine **13** and aldehyde **14** occurred smoothly to yield β-amino alcohol **30** (70% yield, 4.2 g scale, d.r. > 30:1 based on crude ¹H-NMR). At this point, subjection of **30** to HCl in acetone followed by treatment with KOH aqueous solution, promoted several transformations including 5-*exo-trig* cyclization (a possible mechanism is given in the Supplementary Information, page S27) from C23 hydroxyl group onto the C16–C17 alkene, removal of the *N*-sulfinyl group, and lactamization between the primary amine and methyl carboxylic ester, generating the desired product **31** in 72% yield. Remarkably, this bis-cyclization process allowed us to construct the E/F rings by forging a C−O bond and a C−N bond (light blue bonds in **31**) in one pot. With the hexacyclic framework of jervanine-type alkaloids established, reduction of lactam **31** with LiAlH₄ led to (−)-cyclopamine (**4**) at gram-scale (1.40 g), the spectroscopic data of which were in good agreement with those reported[23,24].

### Substrate scope of the biomimetic rearrangement

After completion of the syntheses of veratramine and cyclopamine, we further examined the generality of the key biomimetic rearrangement of steroid diol **32** containing C17-OH to access C-*nor*-D-*homo* steroid **33**. As shown in Fig. 3, subjecting **18** with an acetylenyl group at C17 to the conditions of Tf₂O and pyridine induced the desired rearrangement to provide epoxide **33a** in 84% yield. This method was compatible with silyl ether (**33b**), tertiary amine (**33c**), epoxide (**33d** and **33e**), enone (**33f**), and tertiary alcohol (**33g**). Substrates derived from varied steroid skeletons were also suitable for this protocol, which furnished the rearranged products **33h** (73% yield) and **33i** (82% yield) with good efficiency. In addition, different substituent groups at the C17 position were investigated. As a result, vinyl, cyano, and pyridyl proved to be viable units as demonstrated by the formation of **33j–l** in 52–73% yields. Collectively, this biomimetic rearrangement enabled the production of diverse C-*nor*-D-*homo* steroids under simple and mild reaction conditions. In particular, an epoxy functionality was formed in the process, which could be employed in diverse transformations as showcased in the above synthesis.

### Discussion

To summarize, we have reported by far the most efficient and divergent synthetic approach to (−)-veratramine (**1**, 13-step LLS, 11% overall yield (OY); 15 total steps, 6.3% OY) and (−)-cyclopamine (**4**, 13-step LLS, 6.2% OY; 15 total steps, 3.6% OY) from commercially available and

inexpensive starting materials. We demonstrated the practicability of our synthetic route by conducting all steps on gram scales, which delivered 1.79 g of **1** and 1.40 g of **4** in one batch. The present efficient synthesis strategically relies on a 1,2-alkyl shift to form the C-*nor*-D-*homo* steroid core, a divergent conversion of an epoxy moiety into arene and conjugated diene, and a stereoselective reductive coupling/(bis-)cyclization sequence to build (E)/F-ring system. Notably, the biomimetic rearrangement method tolerates different steroid structures and functionalities, which, in combination with the late-stage construction of the azacycle unit, would facilitate access to diverse derivatives. Furthermore, while currently, the industrial production of almost all steroid drugs relies on semi-synthesis, this study again proves how such an approach would expedite the preparation of complex and bioactive steroids[49–51].

## Methods

All reactions that require anhydrous conditions were performed in flame-dried glassware under Ar atmosphere and all reagents were purchased from commercial suppliers. Dry toluene was obtained according to *Purification of Laboratory Chemicals* (Peerrin, D. D.; Armarego, W. L. and Perrins, D. R., Pergamon Press: Oxford, 1980). Other dry solvents were purchased from Energy Chemical. Reactions were monitored by thin-layer chromatography (TLC) supplied by Yantai Chemicals. Visualization was accomplished with UV light, exposure to iodine, and stained with ethanolic solution of phosphomolybdic acid or basic solution of $KMnO_4$. The reaction products were purified by column chromatography on silica gel (200–300 meshes) from the Anhui Liangchen Silicon Material Company. $^1H$ NMR and $^{13}C$ NMR spectra were recorded on Varian INOVA-400/54 and Agilent DD2-600/54 instruments. The solvent signal was used as reference for $^1H$ NMR ($CD_2Cl_2$, 5.32 ppm, $CDCl_3$, 7.26 ppm, $CD_3OD$, 3.31 ppm, $CD_3CN$, 1.9 ppm, $C_5D_5N$, 7.20, 7.57, 8.72 ppm) and $^{13}C$ NMR ($CD_2Cl_2$, 53.8 ppm, $CDCl_3$, 77.0 ppm, $CD_3OD$, 49.0 ppm, $CD_3CN$, 1.32, 118.3 ppm, $C_5D_5N$, 123.4, 135.4, 149.8 ppm). The following abbreviations are used to explain the multiplicities: $s$ = singlet, $d$ = doublet, $t$ = triplet, $q$ = quartet, br = broad, $m$ = multiplet, and coupling constants ($J$) are reported in Hertz (Hz). Infrared (IR) spectra were recorded on a Perkin Elmer Spectrum Two FT-IR spectrometer. High-resolution mass spectra (HRMS) were recorded on Bruker Apex IV FTMS or Thermo Scientific LTQ Orbitrap XL ESI mass spectrometers. LC-MS analysis was performed on HP Agilent 6420 Triple Quad LC/MS. The specific optical rotation was obtained from Rudolph Research Analytical Autopol VI automatic polarimeter.

## Data availability

Additional data supporting the findings described in this paper are available in the Supplementary Information. Crystallographic data for the structures reported in the present article have been deposited at the Cambridge Crystallographic Data Center (CCDC), under deposition nos. CCDC 2321471 (**15**), 2321472 (**20′**), and 2321473 (**32d**) (see 'X-ray crystallographic data' in Supplementary Information). Copies of the data can be obtained free of charge via https://www.ccdc.cam.ac.uk/structures. All data are available from the corresponding authors upon request.

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

## Acknowledgements

Financial support was provided by the National Natural Science Foundation of China (82222065 to X.Y.L., 22331008 to Y.Q., and 21921002 to Y.Q.). We thank Dr. Wanshu Wang and Dr. Linxi Wan (Sichuan University) for recording NMR and HRMS, respectively. This paper is dedicated to Professor Feng-Peng Wang on the occasion of his 80th birthday.

## Author contributions

W.H. performed the main experiments including methodology development and syntheses of natural products. H.L. and Y.W. investigated the scope of the biomimetic rearrangement. C.L. and J.C. carried out some scale-up experiments. X.Y.L. and Y.Q. designed and supervised the project, and wrote the manuscript with feedback from other authors.

## Competing interests

Y.Q., X.Y.L., and W.H. are listed as inventors on patents 'Synthesis of veratramine' (CN2023118206686) and 'Synthesis of cyclopamine'(CN2024103595370) filed by Sichuan University on some aspects of the work in this paper. The other authors declare no competing interests.
