## [Peer Review File · Nature Communications]

Divergent and gram-scale syntheses of (–)-veratramine and (–)-cyclopamineREVIEWER COMMENTS

Reviewer #1 (Remarks to the Author):

This manuscript, by Qin and co-workers, reports the asymmetric semi-synthesis of C-nor-D-homo-steroidal alkaloids, cyclopamine and veratramine in 13 LLS steps with 11% and 6.2% overall, respectively. A biomimetic rearrangement was employed to form the C-nor-D-homo steroid core, while a stereoselective reductive coupling/(bis-)cyclization sequence was utilized to establish the (E)/F-ring moiety. In my opinion, this work is remarkable for the following aspects: 1) detailed summaries of the published works are provided in Figure 1b. 2) pre-installation of a C17 hydroxyl group would not only facilitate the rearrangement, but also act as a proper functionality to form the anticipated arene D ring or cyclohexadiene D ring, 3) synthetic route by conducting all steps on gram-scale. And, the manuscript was well written and organized. In summary, I think this work is of interest to the broad readership of the Nature Communications.

Suggestions:

- 1) The paper described the gram-scale synthesis of natural products and attempted to elucidate the efficiency of the route. However, the SmI₂ reagent used in the reductive coupling is not cost-effective and difficult to prepare and store, which greatly affects large-scale preparation. More comments or potential solutions should be added in the revised manuscript.
- 2) the diastereoselectivity of the key SmI₂-promoted reductive coupling is stereospecific. This reviewer suggests the authors to recheck the d.r. data with the crude NMR of the reaction. And, more comments, stereo-analysis and a transition state should be added in the manuscript.
- 2) Actually, veratramine is commercially available at a price of \$117.3/gram. This information should be considered and mentioned in the text.
- 3) On page 4, line 68, the font of "tetra" should be italic.

Reviewer #2 (Remarks to the Author):

The manuscript presents an efficient and practical synthesis of both veratramine and cyclopamine based on a divergent approach, addressing the demand for reliable chemical synthesis methods for these valuable compounds. Isosteroidal alkaloids, including veratramine and cyclopamine, hold significant importance because of their various biological activities. However, challenges in their extraction from plants necessitate efficient chemical synthesis methods. Previous syntheses of these alkaloids required numerous steps or yielded limited quantities, underscoring the necessity for innovative and scalable synthetic approaches to meet the demand for these compounds.

Following a similar strategy to the first semi-synthesis of cyclopamine reported by Giannis' group, this work utilized a biomimetic rearrangement to form the C-nor-D-homo steroid core, which commenced with the preparation compound 17 from dehydro-epi-androsterone (DHEA) via a Cu-mediated C-H oxidation protocol. The novelty of this work lies in conducting the 1,2-migration in the presence of a free C17 hydroxyl group, yielding epoxide 15 that could be diverted to two target natural products. The robustness and utility of this rearrangement on steroid diols are further demonstrated by Table 1.

It is noteworthy that pre-installation of the 1-cyanovinyl fragment on C17 significantly facilitated the control of regio- and stereoselectivity during the synthesis. The highly diastereoselective hydrogenations (19 \diamond 20, and 15 \diamond 24) to secure the correct C20 stereogenic center are remarkable. Are there any stereochemical rationales for them? Page 4, last paragraph: "subsequent 1,2-proton shift" should be "subsequent 1,2-hydride shift".

Subsequently, the intermolecular reductive coupling of the chiral tert-butylsulfonamides (12 or 13) and aldehyde 14 were carried out, effectively constructing the F-ring via lactamization. Given the C20 stereogenic center, is this coupling a matched or unmatched case? The cationic cyclization to forge the C17 stereogenic center (30 \diamond 31) is impressive, but a detailed explanation of the mechanism of selectivity is needed.

The authors obtained 1.79 g of (–)-veratramine and 1.40 g of (–)-cyclopamine using their efficient synthetic approach, which should not be perceived as straightforward as they appear due to the delicate and complicated framework. The supplementary supporting information is in a good shape. This work not only has the potential to expedite the preparation of complex and bioactive steroids for functional studies but also raises several interesting stereochemical observations, which could inform future work on complex natural products. Consequently, the reviewer supports the publication of this work in Nature Communications after the minor revision.

Reviewer #3 (Remarks to the Author):

In this manuscript, Dr. Qin, Dr. Liu and co-workers reported their semi-synthesis of veratramine and cyclopamine from DHEA. Both veratramine and cyclopamine are complex and important natural products with biological importance. While these two natural products have been made previously, the current synthesis is novel and important. Strategically, they used DHEA as the starting point which is cheap and readily available and a biomimetic rearrangement to generate the desired C-nor-D-homo skeleton. While a similar strategy has been used in the Ginnanis synthesis of cyclopamine, these two differ significantly in the detail. In this work, two conditions were developed to rearrange 16 to either 15 or 19, which set up the stage for their divergent total synthesis of veratramine and cyclopamine. The authors also explored the substrate scope for the biomimetic rearrangement. To install the piperidine ring, the authors developed a remarkably efficient SmI₂-mediated aldehyde-imine reductive coupling using the Ellman auxiliary to control the stereoselectivity, which is a highlight for this synthesis. In addition, this new synthesis is scalable and resulted both natural products in gram scale. This is important for supporting analog synthesis and biological evaluations. Overall, the manuscript is well-written and is recommended for publication after minor revision.

1. The authors should point out the extreme caution for using Zn(CN)₂ in the experimental procedure.
2. Provide an explanation/model to account for the catalytic hydrogenation with Wilkinson's catalyst as well as the SmI₂-mediated aldehyde-imine reductive coupling.
3. Table S2 and S4, chemical shift differences between the synthetic and natural samples should be given.

Manuscript Title: " Divergent and gram-scale syntheses of (-)-veratramine and (-)-cyclopamine "

Manuscript number: NCOMMS-24-20326A

Point-by-point response to the reviewers' comments

Response to the Comments of Reviewer #1:

*** This manuscript, by Qin and co-workers, reports the asymmetric semi-synthesis of C-nor-D-homo-steroidal alkaloids, cyclopamine and veratramine in 13 LLS steps with 11% and 6.2% overall, respectively. A biomimetic rearrangement was employed to form the C-nor-D-homo steroid core, while a stereoselective reductive coupling/(bis-)cyclization sequence was utilized to establish the (E)/F-ring moiety. In my opinion, this work is remarkable for the following aspects: 1) detailed summaries of the published works are provided in Figure 1b. 2) pre-installation of a C17 hydroxyl group would not only facilitate the rearrangement, but also act as a proper functionality to form the anticipated arene D ring or cyclohexadiene D ring, 3) synthetic route by conducting all steps on gram-scale. And, the manuscript was well written and organized. In summary, I think this work is of interest to the broad readership of the Nature Communications.

Response: We appreciate the reviewer's positive and insightful comments on our manuscript.

***1) The paper described the gram-scale synthesis of natural products and attempted to elucidate the efficiency of the route. However, the SmI₂ reagent used in the reductive coupling is not cost-effective and difficult to prepare and store, which greatly affects large-scale preparation. More comments or potential solutions should be added in the revised manuscript.

Response: The SmI₂ solution employed in the reductive coupling could be purchased from a commercial source or prepared from Sm and I₂ using a well-known procedure. However, this reagent was used as diluted solution (0.1 M in THF) and also difficult to store, which, as the reviewer mentioned, is not very useful for industry scale preparation. According to the suggestions, we added a note on the SmI₂ reagent and a reference for its preparation (ref. 4 in SI) in the revised Supplementary Information. Please see page S14: "*Note: The SmI₂ solution (0.1 M in THF) employed in the reductive coupling could be purchased from a commercial source or prepared from Sm and I₂ using a well-known procedure.⁴ When purchasing on large scale (> 250 g), the price of samarium metal granules is around \$0.15/gram from Huizhou Boguan Vacuum Applied Materials Co., Ltd. (<http://www.bgvmat.com/>, accessed on May 15, 2024).*"

*** 2) the diastereoselectivity of the key SmI₂-promoted reductive coupling is stereospecific. This reviewer suggests the authors to recheck the d.r. data with the crude NMR of the reaction. And, more comments, stereo-analysis and a transition state should be added in the manuscript.

Response: Thanks for the reviewer's suggestion. We double-checked the crude NMR, combined with LC-MS analysis, of the key SmI₂-promoted reductive couplings (**12** + **14** to **28**; **13** + **14** to **30**) and did not find any diastereomer for the two coupling reactions. The crude NMRs are provided as below.

Crude ¹H-NMR for the reductive coupling of **12** and **14**:

Crude ¹H-NMR for the reductive coupling of **13** and **14**:

It should be pointed out that a byproduct (compound **30'**) was isolated in 2.2% yield in the coupling of **13** and **14** reported in our submitted manuscript, which was originally determined as the diastereomer (at C22 and/or C23) formed in the coupling. In our further work aiming at synthesis of the unnatural derivatives of cycloamine, we prepared 20-*epi*-**13** from compound **S5** following the similar synthetic route from **24** to **13**. Subjecting 20-*epi*-**13** and aldehyde **14** to the reductive coupling conditions generated a product, which was found to be the same with compound **30'**. Thus, the isolated byproduct **30'** in the coupling of **13** and **14** was the C20-epimer of **30**, rather than the previously assigned diastereomer at C22 and/or C23. Formation of 20-*epi*-**30** (that is, the correct structure of **30'**) could be due to partial epimerization of the C20 stereocenter at the aldehyde (in **26**) or imine (in **13**) stage.

Based on the above-mentioned results, we first added in the revised manuscript (pages 6 and 7) the description “(d.r. > 30:1 based on crude ¹H-NMR)” for the synthesis of **28** and **30**, respectively. In addition, we added the experimental procedures, characterization data, and NMR spectra for the transformation from compounds **S5** to **30'**, as well as corrected the structure of **30'**. Please see page S25–S27 and page S76–S79 in the revised Supplementary Information.

Concerning the stereocontrol of the SmI₂-promoted reductive coupling, we proposed a five/four-membered bicyclic transition state (see below) in which Sm could coordinate to nitrogen and sulfinyl oxygen of imine. When (*R*)-*N*-*tert*-butanesulfinyl imine was employed, the *Si*-face addition would be favored to yield the (*S*)-amine as the predominant product. According to the reviewer's suggestion, the transition state for the reductive coupling is added in the revised Supplementary Information (page S14), with a note in the main text (page 6).

*** 3) Actually, veratramine is commercially available at a price of \$117.3/gram. This information should be considered and mentioned in the text.

Response: Thanks for the reviewer's suggestion. This information is added in the revised manuscript accordingly. Please see ref. 25: “Veratramine is commercially available (ca. \$120/gram) from Bide Pharmatech Co., Ltd., China (www.bidepharm.com).”

*** 4) On page 4, line 68, the font of “tetra” should be italic.

Response: It has been corrected in the revised version accordingly.

Response to the Comments of Reviewer #2:

*** The manuscript presents an efficient and practical synthesis of both veratramine and cycloamine based on a divergent approach, addressing the demand for reliable chemical synthesis methods for these valuable compounds. Isosteroidal alkaloids, including veratramine and cycloamine, hold significant importance because of their various biological activities. However, challenges in their extraction from plants necessitate efficient chemical synthesis methods. Previous syntheses of these alkaloids required numerous steps or yielded limited quantities, underscoring the necessity for innovative and scalable synthetic approaches to meet the demand for

these compounds.

Following a similar strategy to the first semi-synthesis of cycloamine reported by Giannis' group, this work utilized a biomimetic rearrangement to form the C-nor-D-homo steroid core, which commenced with the preparation compound **17** from dehydro-epi-androsterone (DHEA) via a Cu-mediated C-H oxidation protocol. The novelty of this work lies in conducting the 1,2-migration in the presence of a free C17 hydroxyl group, yielding epoxide **15** that could be diverted to two target natural products. The robustness and utility of this rearrangement on steroid diols are further demonstrated by Table 1.

Response: We appreciate the reviewer's positive and insightful comments on our manuscript.

*** 1) It is noteworthy that pre-installation of the 1-cyanovinyl fragment on C17 significantly facilitated the control of regio- and stereoselectivity during the synthesis. The highly diastereoselective hydrogenations (**19**→**20**, and **15**→**24**) to secure the correct C20 stereogenic center are remarkable. Are there any stereochemical rationales for them?

Response: The diastereoselectivity in transformations of **19** to **20** and **15** to **24** could arise from directed hydrogenations through coordination of the rhodium catalyst with β -oriented O-atom of the hydroxyl group in **19** and of the epoxide group in **15**, respectively, to generate the correct C20 stereocenter.

According to the reviewer's comments, we added the following discussion in the revised manuscript (page 6): "Of note, the directed hydrogenation through coordination of the rhodium catalyst with β -oriented O-atom of the hydroxyl group in **19** secured the correct C20 stereocenter⁴⁷." A closely related literature on the directed hydrogenation has been added as ref. 47 and numbering of the other references has been updated as well. In addition, for the conversion of **15** to **24**, we revised "directly subjecting alkene **15** to catalytic hydrogenation conditions" to "subjecting alkene **15** to directed hydrogenation conditions".

*** 2) Page 4, last paragraph: "subsequent 1,2-proton shift" should be "subsequent 1,2-hydride shift".

Response: It has been corrected in the revised version accordingly.

*** 3) Subsequently, the intermolecular reductive coupling of the chiral tert-butylsulfinamides (**12** or **13**) and aldehyde **14** were carried out, effectively constructing the F-ring via lactamization. Given the C20 stereogenic center, is this coupling a matched or unmatched case? The cationic cyclization to forge the C17 stereogenic center (**30**→**31**) is impressive, but a detailed explanation of the mechanism of selectivity is needed.

Response: The stereochemistry of the intermolecular reductive coupling is controlled by the Ellman auxiliary but not related to the C20 stereogenic center, because we were able to access the products with the same 22*S* and 23*R* configurations when using (*R*)-*N*-tert-butanesulfinyl imines bearing different C20 configurations. Please see page S25 for the conversion of compounds **S5** into **30'** in the revised Supplementary Information.

As for the selectivity of the cationic cyclization to forge the C17 stereogenic center (**30**→**31**), there are two possible transition states for the 5-*exo-trig* cyclization: **TS-1** and **TS-2** (shown below). The former is supposed to be disfavored because it would need to overcome a high reaction energy barrier by placing the large R' group at the axial position. Thus, it is more likely that the reaction proceeds through **TS-2**, which yielded intermediate **A** with the desired C17 stereogenic center and finally led to the product **31**. According to the reviewer's suggestion, we have added the mechanism in the revised SI (page S27) with a note in the main text (page 7): "(a possible mechanism is given in the Supplementary Information, page S27)".

*** 4) The authors obtained 1.79 g of (–)-veratramine and 1.40 g of (–)-cyclopamine using their efficient synthetic approach, which should not be perceived as straightforward as they appear due to the delicate and complicated framework. The supplementary supporting information is in a good shape. This work not only has the potential to expedite the preparation of complex and bioactive steroids for functional studies but also raises several interesting stereochemical observations, which could inform future work on complex natural products. Consequently, the reviewer supports the publication of this work in Nature Communications after the minor revision.

Response: We appreciate the reviewer’s positive and insightful comments on our manuscript.

Response to the Comments of Reviewer #3:

*** In this manuscript, Dr. Qin, Dr. Liu and co-workers reported their semi-synthesis of veratramine and cyclopamine from DHEA. Both veratramine and cyclopamine are complex and important natural products with biological importance. While these two natural products have been made previously, the current synthesis is novel and important. Strategically, they used DHEA as the starting point which is cheap and readily available and a biomimetic rearrangement to generate the desired C-nor-D-homo skeleton. While a similar strategy has been used in the Ginnanis synthesis of cyclopamine, these two differ significantly in the detail. In this work, two conditions were developed to rearrange 16 to either 15 or 19, which set up the stage for their divergent total synthesis of veratramine and cyclopamine. The authors also explored the substrate scope for the biomimetic rearrangement. To install the piperidine ring, the authors developed a remarkably efficient SmI₂-mediated aldehyde-imine reductive coupling using the Ellman auxiliary to control the stereoselectivity, which is a highlight for this synthesis. In addition, this new synthesis is scalable and resulted both natural products in gram scale. This is important for supporting analog synthesis and biological evaluations. Overall, the manuscript is well-written and is recommended for publication after minor revision.

Response: We appreciate the reviewer’s positive and insightful comments on our manuscript.

*** 1. The authors should point out the extreme caution for using Zn(CN)₂ in the experimental procedure.

Response: Based on the reviewer’s suggestion, we have added a note in the experimental procedure for using Zn(CN)₂. Please see page S5 in the revised Supplementary Information: “*Note: Zn(CN)₂ is toxic and must be handled with extreme caution.*”

*** 2. Provide an explanation/model to account for the catalytic hydrogenation with Wilkinson's catalyst as well as the SmI₂-mediated aldehyde-imine reductive coupling.

Response: The same suggestions have been given by the above two reviewers. Please see our response to Comment 1 of Reviewer #2 and Comment 2 of Reviewer #1, respectively, for the explanation/model to account for the catalytic hydrogenation with Wilkinson's catalyst as well as the SmI₂-mediated aldehyde-imine reductive coupling.

*** 3. Table S2 and S4, chemical shift differences between the synthetic and natural samples should be given.

Response: Modifications have been made according to the suggestion. Please see pages S19 and S31 in the revised Supplementary Information.

For your convenience, all the above-mentioned corrections have been highlighted in yellow backgrounds in the revised manuscript and Supplementary Information.

REVIEWERS' COMMENTS

Reviewer #1 (Remarks to the Author):

The authors have totally address my concerns in the revised manuscript. The revised version is ready for the publication.

Reviewer #2 (Remarks to the Author):

The revision has addressed reviewer's questions and concerns appropriately. Consequently, publication of the revised version on Nature Communications is now strongly recommended.

Reviewer #3 (Remarks to the Author):

The authors addressed the questions well. The manuscript is recommended for publication.